# Genetic Variation of *Puccinia triticina* Populations in Iran from 2010 to 2017 as Revealed by SSR and ISSR Markers

**DOI:** 10.3390/jof9030388

**Published:** 2023-03-22

**Authors:** Zahra Nemati, Ali Dadkhodaie, Reza Mostowfizadeh-Ghalamfarsa, Rahim Mehrabi, Santa Olga Cacciola

**Affiliations:** 1Department of Plant Protection, School of Agriculture, Shiraz University, Shiraz 71441-65186, Iran; 2Department of Plant Production and Genetics, School of Agriculture, Shiraz University, Shiraz 71441-65186, Iran; 3Department of Biotechnology, College of Agriculture, Isfahan University of Technology, Isfahan 81431-53784, Iran; 4Department of Agriculture, Food and Environment (Di3A), University of Catania, 95123 Catania, Italy

**Keywords:** *Basidiomycota*, brown rust, genotypes, ISSR, molecular markers, SSR, virulence, *Poaceae*, wheat leaf rust

## Abstract

*Puccinia triticina* is a major wheat pathogen worldwide. Although Iran is within the Fertile Crescent, which is supposed to be the center of origin of both wheat and *P. triticina*, the knowledge of the genetic variability of local populations of this basidiomycete is limited. We analyzed 12 inter simple sequence repeats (ISSRs) and 18 simple sequence repeats (SSRs) of 175 *P. triticina* isolates sampled between 2010 and 2017 from wheat and other *Poaceae* in 14 provinces of Iran. SSRs revealed more polymorphisms than ISSRs, indicating they were more effective in differentiating *P. triticina* populations. Based on a dissimilarity matrix with a variable mutation rate for SSRs and a Dice coefficient for ISSRs, the isolates were separated into three large groups, each including isolates from diverse geographic origins and hosts. The grouping of SSR genotypes in UPGMA dendrograms was consistent with the grouping inferred from the Bayesian approach. However, isolates with a common origin clustered into separate subgroups within each group. The high proportion of heterozygous alleles suggests that in Iran clonal reproduction prevails over sexual reproduction of the pathogen. A significant correlation was found between SSR and ISSR genotypes and the virulence phenotypes of the isolates, as determined in a previous study.

## 1. Introduction

Leaf rust, also known as brown rust, caused by *Puccinia triticina* Eriksson, is a major disease of wheat (*Triticum aestivum* L. and *T. durum* Desf.) globally [1,2]. *Puccinia triticina* is a heteroecious basidiomycete and requires a telial/uredinial host and an alternative (pycnial/aecial) host to complete its life cycle. It is thought that the center of origin of this pathogen is the Fertile Crescent as in this historical region, also including part of present-day Iran, the natural ranges of the primary (wheat) and the alternate host (*Thalictrum* species) overlap [3]. Although leaf rust has a worldwide distribution, alternate hosts are absent or occur sporadically in most wheat-producing areas, and consequently the sexual stage of *P. triticina* has a secondary epidemiological role globally [4]. As a matter of fact, several lines of evidence indicate that asexual reproduction of this pathogen prevails on a world scale [5,6]. Urediniospores of *P. triticina* are wind-dispersed even over long distances, which can result in the intercontinental spread of new genotypes of the pathogen and the consequent outbreaks of wheat leaf rust epidemics [7]. Despite the lack of alternate hosts, the populations of the leaf rust pathogen show a high variability. New genotypes and breeds of *P. triticina* may also emerge in local populations of the pathogen in the absence of sexual recombination. This may occur because of mutations, somatic recombination, the selective effect of large-scale cultivation of wheat cultivars with race-specific resistance genes and migration from distant geographical regions and from other wild or cultivated Poaceae, such as bread wheat, durum wheat, barley and oats, through wind-dispersed urediniospores. [3,8,9,10]. In particular, mutations, resulting in new genetic variants, are likely the primary source of variation in *P. triticina* populations [7]. In wheat-producing countries, large-scale surveys aimed at monitoring the diversity of virulence phenotypes (races) in *P. triticina* populations have been performed repeatedly as the control of leaf rust relies mainly on wheat cultivars with race-specific resistance genes [4,9,11,12,13,14,15,16]. The analysis of the genetic variability of *P. triticina* populations can provide further insight into the biology and ecology of the leaf rust pathogen. The genetic variability of *P. triticina* populations has been investigated in several wheat-production areas worldwide, often in combination with the study of the variability of virulence phenotypes. Different molecular markers have been used, including random amplified polymorphic DNAs (RAPDs), amplified fragment length polymorphisms (AFLPs), inter simple sequence repeats (ISSRs), locus specific microsatellites or simple sequence repeats (SSRs) and single nucleotide polymorphisms (SNPs) [12,17,18,19,20,21,22,23,24]. While RAPDs and AFLPs are dominant, SSR and ISRR markers are co-dominant and can distinguish heterozygous and homozygous genotypes [3,7]. SSR markers unraveled similarities between durum wheat isolates of *P. triticina* from Europe and South America and among populations of the pathogen from diverse host plants or geographical origins [7,9,16,18,25]

A more precise genetic analysis of SSRs data has become possible with a recently developed approach based on different models of SSR evolution [26], which consider the stepwise mutation model (SMM) in an either variable (SMMv) or constant (SMMc) mutation rate scenario [26]. Notably, both the SSM and the infinite alleles model (IAM) apply to SSR multi-locus genotypes and are based on a comparison of allele patterns of SSR loci rather than allele frequencies, which is essential in the case of the association between alleles as it occurs in asexually reproducing organisms [26,27].

Previous studies investigated the variability of virulence phenotypes of *P. triticina* populations in Iran [10,28]. The objective of this study was to explore the genetic variability of these populations from 2010 to 2017 using SSRs and ISSRs as molecular markers.

## 2. Materials and Methods

### 2.1. Puccinia triticina Isolates

Between 2010 and 2017, leaf rust samples were collected from various species of plants in the *Poaceae* family, including bread wheat (*T. aestivum)* durum wheat (*T. durum* Desf.), barley (*Hordeum vulgare* L.), wild barley (*H. vulgare* ssp. *spontaneum* Thell.), oat (*Avena sativa* L.), and triticale (×*Triticosecale* Wittm. ex A. Camus), in wheat fields of 14 provinces of Iran, and single uredinium isolates were obtained from each sample. The virulence phenotype of the set of isolates examined in this study was identified in a previously published study by testing them on 20 differential Thatcher’s near-isogenic wheat lines carrying single-leaf rust resistance genes [10,29] (Appendix A). Leaf rust isolates from the same region and with identical virulence phenotypes were excluded, and 175 single-uredinium leaf rust isolates were selected for molecular genotype analysis, including 146 isolates from bread wheat (*Pt*—bread wheat) (Appendix A).

To propagate the isolates, a single pustule (uredinium) was used to inoculate the susceptible wheat cv Boolani in greenhouse at 20–24 °C according to the method described by Kolmer et al. [30]. Urediniospores of each isolate produced on artificially inoculated wheat plantlets of the cv Boolani were dried in a desiccator and stored at −80 °C for later use.

### 2.2. DNA Extraction

DNA was extracted from 20 mg frozen urediniospores of each single-uredinium isolate using the CTAB method and the DNGTM.PLUS extraction kit (CinnaGen, Tehran, Iran) [31]. The DNA was quantified both with a NanoDrop spectrophotometer (NanoDrop Technologies, Wilmington, DE, USA) and on 0.8% agarose gel in 1X TBE (Tris-Borate-EDTA) buffer (pH 8.3), stained with ethidium bromide (0.25 mL^−1^) and visualized under UV light.

### 2.3. ISSR Markers

The ISSR genotypes of the isolates were determined using a set of 12 ISSR markers (Table 1) and according to the protocol of Spackman et al. [32]. The amplification reactions were performed in a thermal cycler (CG1.96, Corbett Research, Mortlake, NSW, Australia) using the following protocol: initial denaturation at 94 °C for 2 min, 35 cycles of denaturation at 94 °C for 30 s, annealing at 45 °C for 1 min, extension at 72 °C for 1 min, final extension at 72 °C for 5 min. The fragments were separated on 2% agarose gel in TAE stained with 0.1% ethidium bromide. All the images were captured in a gel documentation system (Syngene, Frederick, MD, USA). Fragment size was estimated with a medium-range ruler (50–1500 bp, DM1100, Life Science Products, Stevensville, MD, USA) and the bands with a range of 100–2000 bp were scored visually. To record the molecular data for the markers, a binary value matrix was prepared, where the absence of a band was represented as 0 while the presence of an amplicon was represented as 1. This approach allowed us to perform a statistical analysis of diversity between genotypes and compare them with SSR markers [33,34,35].

### 2.4. SSR Markers

Clonal genotypes, i.e., isolates with identical virulence phenotypes and ISSR genotypes, were excluded and the remaining 126 isolates were genotyped using a set of 18 microsatellite primer pairs developed from the genomic libraries of *P. triticina*, including PtSSR 3, PtSSR 13, PtSSR 50, PtSSR 55, PtSSR 61, PtSSR 76, PtSSR 91, PtSSR 92, PtSSR 151A, PtSSR 152, PtSSR 154, PtSSR 158, PtSSR 161, PtSSR 164, PtSSR 173, PtSSR 186, PtSSR 68.1, and PtSSR 184 [19]. Polymerase chain reaction (PCR) conditions were as described by [19]. Amplifications were conducted in a thermal cycler (CG1.96, Corbett Research, Mortlake, NSW, Australia) using the following temperature profile: initial denaturation step at 98 °C for 30 s, then 30 cycles at 98 °C for 30 s, annealing temperatures for 30 s, and 72 °C for 30 s followed by a final extension step of 72 °C for 10 min. A range of annealing temperatures from 58 to 64 °C was used for each primer pair. PCR products were separated on 7% polyacrylamide gel in 1 × TBE buffer at 100 volts for 150 min. At the end of the run, the gels were stained in silver nitrate (AgNO3) solution (0.1 g 100 mL^−1^) [43]. Allele sizes for each primer were visually determined using DM1100 size standards. DNA bands generated by each primer pair were standardized with the allele sizes in the initial characterization of the SSR primers and other *P. triticina* isolates previously characterized using the same set of SSR primer pairs [19]. Individual isolates were scored for dikaryotic genotypes for each SSR locus by recording allele sizes in base pairs for both alleles at each locus.

### 2.5. Data Analysis

A binary matrix for 8 polymorphic ISSRs and 175 isolates was constructed in Microsoft^®^ Excel. Additionally, a matrix for SSR genotypes (126 isolates) was obtained based on the allele size of each specific microsatellite fragment. The Polymorphism Information Content (PIC) and the number of alleles were calculated for these ISSR and SSR markers. Data analysis was performed using LOCUS software, and single locus parameters for 126 isolates were also calculated [26]. The dissimilarity between ISSR genotypes was measured with either the simple mismatch, Jaccard or Dice indexes [26]. Considering the dikaryotic nature of *P. triticina*, the SSR genotypes were analyzed as diploid organisms. The dissimilarity matrix between SSR genotypes was calculated assuming two different models of microsatellite evolution [26]: the infinite alleles model (IAM) and stepwise mutation model (SMM) with constant mutation rate (SMMc) and SSM with variable mutation rate (SMMv) at different loci [26,27].

By comparing the correlation of virulence phenotype with genotypes, the simple mismatch, Jaccard, and Dice dissimilarities of the IAM model for virulence phenotype and ISSR with IAM and SMM model of SSR were calculated and used. To examine the correlation of dissimilarity matrices, we calculated the Mantel test with different values for each pair of matrices for SSR, ISSR, and virulence phenotype data sets [44]. Mantel tests were calculated using the MXCOMP program of the NTSYS pc package, version 2.2 (Exeter Software, East Setauket, NY, USA). Associations of virulence phenotypes, ISSR and SSR genotypes were estimated for clone-corrected data.

The dissimilarity matrix (SMMv) for SSR and the Dice coefficient for ISSR were used to generate dendrograms, using the unweighted pair group method with arithmetic mean (UPGMA) and neighbor-joining to illustrate the genetic relationships between genotypes. Analyses were performed using the Power Marker V3.25 [45], and the dendrograms were visualized in Mega 7.1 [46].

To test the hypothesis that sexual reproduction contributes to the genetic variability of leaf rust populations in Iran, we used STRUCTURE v. 2.3 to group the SSR genotypes of the isolates with a Bayesian approach [47,48]. This method has been developed for use with sexual populations and assumes the populations to be in Hardy–Weinberg equilibrium. The parameters of the project were set as: (i) admixture model (assuming that individuals may have part of the genome from each of the k populations) and independent allele frequencies among populations; and (ii) run length was given as 100,000 burning period lengths, followed by 100,000 Markov Chain Monte Carlo (MCMC) replications. Values of k (groups) from 1 to 10 were used, with 10 iterations for each level of k. The average ln Pr(X/K) was calculated for each k and the differences between sequential values corresponding to k 1–10 were determined as described in [49]. The number of SSR groups was determined by the k interval with the largest change in ln Pr (X/K).

## 3. Results

Overall, 175 single-uredinium isolates of leaf rust sampled from various hosts in 14 provinces of Iran from 2010 to 2017 (Appendix A) were tested using a set of 12 ISSR primers. However, polymorphic band patterns were obtained only with 8 out of the 12 primers. In total, 126 bands were analyzed among the 175 isolates, and 121 were polymorphic. The proportion of polymorphic bands was variable, with an average of 44.5 ± 8.4% (Table 2). The ISSR primers could not detect polymorphic alleles in oat and triticale isolates. After excluding clones, i.e., isolates from the same region with identical virulence phenotypes and ISSR genotypes, 126 isolates were genotyped using SSR markers.

### SSR and ISSR Polymorphisms

Only 15 out of 18 SSR primers detected polymorphisms among the 126 isolates analyzed. PtSSR 76 and PtSSR 151A primers produced polymorphic band patterns in isolates from wild barley. Nine primer pairs (PtSSR 92, PtSSR 151A, PtSSR 154, PtSSR 158, PtSSR 161, PtSSR 164, PtSSR 173, PtSSR 68.1, and PtSSR 184) detected polymorphisms in isolates from durum wheat. On average, the SSR primers produced 67 bands, of which 55 were polymorphic (Table 2). The mean percentage of polymorphic bands was 75 ± 3.3%. The average PIC values for ISSR and SSR were 0.23 and 0.29, respectively. Data analysis showed that the number of heterozygous alleles per locus was greater than that of homozygous, and the minimum and maximum number of alleles per locus were two and five, respectively (Table 3).

The SSR, ISSR, and virulence matrices were compared for correlation with Mantel tests. There were relatively low but significant correlations between dissimilarity matrices of SSRs (IAM, SMMc and SMMv), dissimilarity matrices of ISSRs and virulence phenotypes (Dice, Jaccard coefficient and simple mismatch) (Table 4).

The highest correlation coefficient was found between SSR (SMMc) and ISSR (Dice) genotypes (r = 0.495, *p* = 0.001).

The correlation between SSR genotypes (SMMv) and virulence phenotypes based on the Dice coefficient (r = 0.287, *p* = 0.001) was nearly identical to that of ISSR genotypes and virulence phenotypes based on the Jaccard coefficient (r = 0.288, *p* = 0.001).

The unweighted dendrograms based on the Power Marker program results are shown in Appendix A (neighbor-joining dendrograms based on Dice for ISSR genotypes) and Appendix A (neighbor-joining based on SMMv for SSR genotypes). Both dendrograms based on SSR and ISSR analysis grouped genotypes into three major clades. However, the grouping of genotypes based on SSRs and ISSRs did not match. Clade I of the neighbor-joining dendrogram of ISSR genotypes encompassed *P. triticina* isolates collected from 2010 to 2012. Clade II encompassed the *P. triticina* isolates collected in 2013 and many isolates collected in 2014. Clade III encompassed multiple subgroups, including isolates from durum wheat, oat, triticale, barley and wild barley, respectively (Appendix A).

As shown in Appendix A, the three major clades obtained with SSRs were not related to the year of collection. In the SSR dendrogram, clade I consisted of barley, oat and triticale isolates collected in 2016 and some bread wheat isolates collected in 2013, 2014, 2015, and 2016. Clade II comprised three subclades. Subclade II-1 grouped part of the bread wheat isolates collected in 2010, 2011, 2012, 2014, 2015 and 2016, as well as durum wheat isolates collected in 2016 in Khuzestan Province. A few isolates from bread wheat collected in 2010, 2011 and 2012 grouped in subclade II-2. Subclade II-3 included isolates from wild barley collected in the Margoon protected region in 2017, durum wheat isolates from Ilam Province collected in 2016, and a few bread isolates collected in 2010, 2011, 2013, 2015 and 2016. Finally, clade III consisted of isolates from wild barley collected in Bamu national park and bread wheat isolates collected in 2011, 2013 and 2014 (Appendix A). Two barley isolates (95-8-1 and 95-17-1) clustered into a small group, outside the three major clades. Grouping SSR genotypes with STRUCTUREv 2.3, the optimal k value (number of clusters) was 3, with a probability of 0.999. A weak grouping was observed among SSR genotypes with k = 5 (Figure 1).

Grouping based on k = 3 showed a good fit to the SSR genotypes data. Therefore, the isolates were placed into three SSR genotype groups based on the change in lnPr (X/K) (Appendix A). The STRUCTUREv 2.3 groups were visualized with colored branches in a neighbor-joining tree of SSR (Appendix A).

Almost all groups of the isolates generated by STRUCTUREv 2.3 conformed to the neighbor-joining groups, assuming the SMMv model of SSR evolution of simple sequence repeats (SSRs) with dissimilarity matrix (Dice, Jaccard coefficient and simple mismatch) of inter simple sequence repeats (ISSs) and virulence phenotypes of *P. triticina* isolates from Iran.

## 4. Discussion

In this study, SSR and ISSR molecular markers were used to examine the genetic variability of a large set of leaf rust isolates from six hosts of the *Poaceae* family sampled in major wheat-producing provinces of Iran for eight consecutive years. This is the first comprehensive study of the genetic structure of leaf rust pathogen populations in Iran and the first report of the application of ISSR markers to the study of its genetic variability. All eight selected SSR loci revealed a considerable polymorphism. The mean proportions of polymorphisms were 75% for SSRs and 44.5% for ISSRs, confirming a higher discriminating power of SSRs as genetic markers. Moreover, ISSRs did not detect polymorphic alleles in isolates from oat and triticale and were less informative than SSRs in revealing allelic variation.

The analysis of the genetic variability of this large collection of isolates using SSR markers showed that the number of heterozygous alleles was by far greater than the number of homozygous alleles, suggesting a prevalence of clonal reproduction in the leaf rust pathogen populations in Iran. This is consistent with the results of similar studies in other wheat-producing areas of the world [50,51,52,53]. SSR primers have been widely used to analyze the genetic variability of *P. triticina* isolates from bread wheat in several wheat-producing areas of the world [19]. Eight distinct SSR groups were characterized in populations of this basidiomycete in Europe, six in North America, and three in China and Pakistan [15,25,50,52]. A recently developed set of SSR primers was used to analyze the genetic variability of *P. triticina* isolates from wheat, triticale and rye [53,54]. In this study, the same SSR markers as those used by [19] to analyze the genetic diversity of *P. triticina* populations from bread wheat were also effective in revealing the genetic variability of the leaf rust pathogen populations from other hosts of the family *Poaceae*, including durum wheat, barley, oat and triticale. A correlation was found between the grouping based on SSR genotypes and the host or geographic area of origin. Based on SSR markers, the isolates from oat and triticale grouped in clade I with isolates from bread wheat collected in various provinces. Conversely, the *P. triticina* isolates from durum wheat shared several genetic similarities with other isolates from bread wheat originating from diverse provinces and clustered with them in clade II. However, barley and wheat isolates formed clearly distinct subgroups. The genetic distance between leaf rust isolates from durum wheat and isolates from oat is not surprising as durum wheat isolates do not infect these two plant species of *Poaceae* (Appendix A) [55]. *Puccinia triticina* isolates from durum wheat collected in two different provinces clustered together in clade II, but into two separate subgroups. This is consistent with previous studies showing that *P. triticina* isolates from durum wheat grouped based on the area of origin [16,56]. Additionally, the isolates from wild barley, all collected in 2017 but originating from diverse geographical areas (Margoon Protected Region and Bamu National Park, two nature reserves in the Fars province), clustered separately in clades II and III, respectively, indicating that these two pathogen populations have a different genetic background despite being collected from the same host. It is likely that these two local populations evolved independently in scarcely anthropized areas characterized by less intensive agriculture and were separated by geographical barriers such as mountains. Moreover, in these areas the proximity between wild barley and spontaneous plants comprising alternate hosts might have conditioned the differentiation process and favored sexual reproduction and genetic recombination. This last hypothesis is being investigated [10]. Conversely, the genetic relatedness between isolates of the leaf rust pathogen from barley and those from wheat could be the consequence of gene flow among populations from different hosts of the same family. Similarly, isolates from oat and triticale clustered together in clade I with isolates from wheat. Triticale (×*Triticosecale*) is an artificial hybrid between wheat (*Triticum* sp.) and rye (*Secale* sp.), whose resistance to leaf rust is likely inherited only by genes derived from the wheat parent [55,57]. The virulence reaction of *P. triticina* isolates from triticale on bread wheat, and vice versa the avirulence response of wheat isolates on triticale, would support this hypothesis [55]. Because of its hybrid origin, triticale could be a bridge enabling the transfer of new leaf rust variants from wheat to rye and vice versa. A recent study of the diversity of virulence phenotypes of the leaf rust pathogen in Iran reported that populations from barley, durum wheat, oat, triticale and wild barley were different from those of bread wheat. This would suggest that cross-infections between bread wheat populations of *P. triticina* and populations of leaf rust from other *Poaceae* are relatively rare events [10]. In contrast with this hypothesis, the same authors found that the virulence phenotype of isolates from barley and wild barley was very similar to the virulence phenotype of bread wheat isolates. Both hypotheses are consistent with the results of the present study, which showed that the SSR genotypes of isolates from barley and wild barley were distinct from those of isolates from wheat but varied greatly and clustered into different clades, also encompassing isolates from wheat and other species of *Poaceae*. Overall, results of the genetic analysis of leaf rust isolates indicate that populations from barley, wild barley, oat and triticale, although distinct from populations associated to bread wheat, have numerous genes in common with them, indicating a gene flow between populations associated to different hosts. It can be speculated that populations of leaf rust associated to barley, wild barley, oat, rye, triticale and perhaps other spontaneous species of *Poaceae* may be a source of variation for populations of *P. triticina* associated to wheat. This would have relevant epidemiological implications as the leaf rust fungus reproduces prevalently or almost exclusively clonally over winter or survives over summer on weeds or other crops of *Poaceae*. Globally, the most common causal agents of rust of barley and oat are *P. hordei* and *P. coronata* f. sp. *avenae,* respectively [58,59]. Barley is also reported as an occasional host of *P. triticina* [2,60,61]. Interestingly, in this study isolates obtained from barley and oat infected and could be propagated on bread wheat cv Boolani, confirming previous reports [10,55]. Based on these results, it can be inferred that barley and oat may be regarded as marginal hosts of *P. triticina* and their resistance to this pathogen is of the near-NHR (nonhost resistance) level [62,63,64], as all bread wheat cultivars we tested, except the cv Boolani, were immune to the isolates from these two cereal crops. In this respect, it would be interesting to further characterize the isolates of leaf rust from barley and oat included in this study.

As differences in the size of each set of isolates, grouped on the basis of host, sampling year, geographical area of origin or virulence phenotype, may affect the analysis of the genetic structure of populations, for the analysis of SSRs and ISSRs data we used LOCUS software, which assumes different models of evolution for these markers [26]. Notably, the recently developed methods for SMM and IAM are based on the comparison between allele patterns rather than allele frequencies of SSR loci, which is relevant in the case of the association between alleles as it occurs in clonally reproducing organisms.

The overall results of this analysis provide circumstantial evidence that mutation, migration, and gene flow are major evolutionary events shaping the populations of the leaf rust pathogen in Iran. In particular, the high proportion of heterozygous alleles revealed by SSRs is consistent with the assumption that clonal reproduction prevails in these populations. In fact, it is assumed that even if sexual reproduction contributes to increasing heterozygosity, clonal reproduction is essential to maintain it in a population. The SSR and ISSR markers differed in their ability to detect genetic polymorphisms in *P. triticina* and provided different information on the genetic structure of populations. The genetic structure of the leaf rust pathogen populations in Iran, as determined by the analysis of ISRRs, correlated quite well with the grouping of isolates based on both the year of sampling and host of origin and was also consistent with their grouping based on virulence phenotype, as characterized in a recent study [10]. Conversely, the three major clades evidenced using SRR markers were not related to annual or host populations. Very probably, SSRs exhibited a considerable admixture of clusters of genotypes because dissimilarity based on SMM reflects genetic differences between individuals.

The two clustering methods used for analyzing the data obtained with SSRs (neighbor-joining and STRUCTUREv 2.3) produced almost consistent results. This is in agreement with a previous study that compared the two methods for characterizing the genetic diversity of *P. triticina* populations in Central Asia and Caucasus [7]. STRUCTUREv 2.3 analysis assigns individuals to a population that represents the best fit for the variation patterns found and infers the origins of individuals when population admixture has occurred [65].

A recent study of the diversity of virulence phenotypes in populations of the leaf rust fungus in Iran demonstrated that new genotypes are mainly introduced from neighboring and even distant countries through wind-dispersed urediniospores [10]. Based on the distribution and prevalence of diverse pathotypes, the route of these introductions was traced back and the dominant winds were identified as the main driving factors [10]. Furthermore, in the cited study, evidence was provided that the deployment of wheat cultivars with race-specific resistance genes has contributed to the selection of new pathotypes of *P. triticina* in Iran.

Iran is within the Fertile Crescent macroregion, where the natural range of the primary (*Triticum* species) and alternate (*Thalictrum* species) hosts of *P. triticina* overlap [1,66,67]. Despite this, no evidence has so far been provided that sexual reproduction contributes to increasing the genetic variability of *P. triticina* populations in this country. To this end, research should focus on less disturbed areas, such as parks or nature reserves, where wheat and related ancestral species coexist with potential alternate hosts.

## Figures and Tables

**Figure 1 jof-09-00388-f001:**
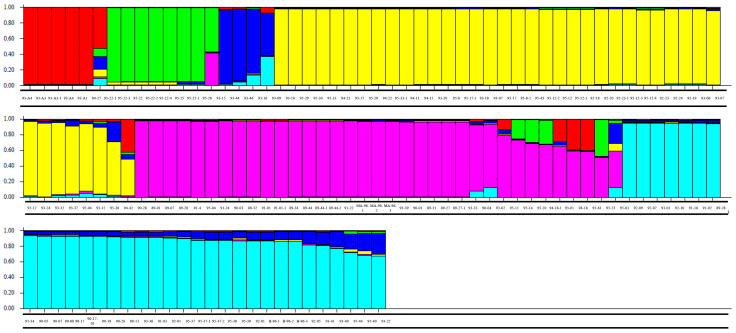
Bayesian assignment of population structure for the genotypes of the leaf rust pathogen. STRUCTURE bar plot of the estimated membership coefficient for each individual for k = 5. Each color indicates a distinct population.

**Table 1 jof-09-00388-t001:** Inter simple sequence repeat (ISSR) primers tested to study the genetic diversity of leaf rust isolates from Iran.

Primer	Primer Sequence	Reference
P1 *	5′-(GACA) 4AT-3′	[36]
P2 *	5′-(GACC) 4TT-3′	[36]
P5 *	5′-(ACTG) 3ACG-3′	[37]
P7	5′-(TGTC) 5-3′	[38]
P8	5′-(CAC) 5-3′	[39]
P9	5′-(CCA) 5-3′	[39]
P10 *	5′-(GACC) 4-3′	[39]
P17 *	5′-(GGAGA) 3-3′	[40]
M1 *	5′-(ACTG) 4-3′	[41]
M2 *	5′-(GACAC) 3-3′	[41]
M6	5′-(GCC) 3-3′	[42]
M7 *	5′-(GAG) 5-3′	[42]

* The ISSR primers used to unravel the genetic diversity of leaf rust isolates from Iran. Primers that are not marked with an asterisk failed in amplifying the corresponding sequences.

**Table 2 jof-09-00388-t002:** Proportion of polymorphic loci as revealed using simple sequence repeats (SSRs) and inter simple sequence repeats (ISSRs) as molecular markers in leaf rust isolates collected in Iran. Isolates are grouped according with the sampling year and host of origin.

Groups of Isolates	SSR Markers	ISSR Markers
Wild Barley 2017	76.47%	2.38%
Durum Wheat 2016	58.82%	34.92%
Barley 2016	70.59%	44.44%
Oat 2016	58.82%	0.00%
Triticale 2016	58.82%	0.00%
Bread Wheat 2016	88.24%	74.60%
Bread Wheat 2015	76.47%	50.79%
Bread Wheat 2014	94.12%	75.40%
Bread Wheat 2013	82.35%	65.08%
Bread Wheat 2012	76.47%	65.87%
Bread Wheat 2011	82.35%	69.84%
Bread Wheat 2010	76.47%	51.59%
Mean	75.00%	44.58%
SE	3.33%	8.39%

**Table 3 jof-09-00388-t003:** Descriptive parameters of SSRs data for *Puccinia triticina* genotypes.

Locus Number	Repeat Size	Missing Data	Min	Max	Number of Homozygotes	Number of Heterozygotes	Proportion of Homozygotes	N. Alleles
1	2	0	128	130	0	126	0	2
2	3	0	360	366	3	123	0.024	3
3	2	0	302	304	51	75	0.405	2
4	3	0	297	303	0	126	0	3
5	4	0	319	327	35	91	0.278	4
6	5	0	307	317	33	93	0.262	5
7	3	0	393	402	0	126	0	3
8	2	0	348	380	11	115	0.087	2
9	3	0	242	252	8	118	0.063	3
10	2	0	470	476	118	8	0.937	2
11	2	0	384	388	47	79	0.373	2
12	4	0	265	272	0	126	0	4
13	3	0	242	250	0	126	0	3
14	2	0	227	232	1	125	0.008	3
15	3	0	213	217	15	111	0.119	3
16	5	0	214	224	11	115	0.087	5
17	4	0	211	220	0	126	0	4
18	4	0	345	400	0	126	0	4
19	4	0	340	347	22	104	0.175	3
20	3	0	335	337	0	126	0	3

**Table 4 jof-09-00388-t004:** Correlation coefficients between dissimilarity matrix (IAM, SMMc and SMMv).

Matrix Correlation	IAM SSR	SMMc SSR	SMMv SSR	DICE ISSR	JACCARD ISSR	MIS MATCH ISSR	DICE Phenotype	JACCARD Phenotype	MIS MATCH Phenotype
IAM SSR	1								
SMMc SSR	0.9187	1							
SMMv SSR	0.92277	0.99452	1						
DICE ISSR	0.3328	0.49538	0.47167	1					
JACCARD ISSR	0.34361	0.49409	0.47025	0.99814	1				
MIS MATCH ISSR	0.28984	0.44524	0.43044	0.95371	0.95843	1			
DICE Phenotype	0.10095	0.26786	0.28764	0.28271	0.28888	0.21861	1		
JACCARD Phenotype	0.10095	0.26786	0.28764	0.28271	0.28888	0.21861	1	1	
MIS MATCH Phenotype	0.05111	0.18975	0.18438	0.19746	0.20227	0.14982	0.91504	0.91504	1

## Data Availability

Data are available upon request.

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
