# Peer review of "Genetic Variation of Puccinia triticina Populations in Iran from 2010 to 2017 as Revealed by SSR and ISSR Markers"

_jof, 2023, doi:10.3390/jof9030388_

Round 1
Reviewer 1 Report
The authors report the genetic variation between 175 isolates of the wheat pathogen Puccinia triticina in Iran. The experimental methodology based on ISSRs and SSRs is appropriate and indicated the latter show more variation. Three major clades were identified and all SSR loci displayed significant polymorphism. While similar studies have appeared elsewhere, this is the first from Iran and suitable for publication.
Author Response
The authors thank the reviewer for the postivie comments.
Kind regards
S. Olga Cacciola
Reviewer 2 Report
Dear Editor,
This study focused on a worldwide important fungal pathogen, Puccinia triticina. The authors analyzed the Iran population from six Poaceae hosts via ISSR and SSR. I think that these strains are precious and the authors made great efforts to collect them over a long time, however, they did not catch the key point and wasted too much space to compare ISSR and SSR. I suggest listing the phylogenetic tree of all isolates and making a comprehensive comparison among populations from different lineages or hosts. In addition, some analysis methods were not applied correctly, e.g. He and STRUCTURE. I recommend a major revision.
Line 24, isolates with ... within each group. What supported this and what did this result mean?
Line 25 It seems the authors want to test the reproductive mode of Iran population. However, we can not make a conclusion only based on high heterozygous alleles. Please calculate observed heterogeneity (Ho) and expected heterogeneity (He), it’s revealed a clonal population if Ho is larger than He. DOI: 10.1007/s10592-020-01262-w
Line 88-96 Please adjust the line spacing
Line 131-146 move the line number to the most right
As I know, the distinguishability of ISSR is lower than SRR, why not amplify the isolates by SSR directly?
Fig.1 Please label the isolate's name
Line 98. I did not see Table S1 and not sure how many isolates were collected from each host. I think these strains are precious and should be the highlight point of your study. I suggest comparing the population genetic structure of pt isolates from different hosts and reducing words on comparable of SSR and ISSR.
Line 191 I don’t think that the STRUCTURE results can answer the existence of sexual reproduction. You should calculate IA or run PHI test in Splitree
Line 216 What’s the meaning of subtitle 3.1?
Table 3. Delete the column of Locus number and put the primer name instead.
Table 4. In my view, all the analyses and results of the correlation between SSR and ISSR are meaningless. For rust species, a dikaryotic organism, the dominant marker of ISSR is incompetent. I would like to analyze the population by SSR directly and put more words on the variation among hosts and other scientific questions.
Line 250 I did not see the supplementary. Maybe I miss it somewhere. I suggested moving Fig.S1 and S2 to the main text. The phylogenetic relationship of these strains is an important result.
Author Response
See the attached response letter
Kind regards
S. Olga Cacciola

Reviewer 3 Report
This manuscript describes a variability of P. triticina from common wheat and other Poaceae hosts present in Iran with SSR and ISSR molecular markers. Based on the dissimilarity matrix isolates divided into 3 groups. (253-261) “Clade I consisted of barley, oat, triticale and some bread wheat isolates. Clade II comprised three subclades. Subclade II-1 grouped part of the bread and durum wheat isolates, II-2 - bread wheat isolates, subclade II-3 included isolates from wild barley, durum wheat and bread isolates. Clade III consisted of isolates from wild barley and bread wheat isolates.
It is known that caused agent of oat rust is Puccinia coronata, and barley rust is Puccinia hordei. The main point of contention for me concerns the host plants. In pp 99-100 authors write "To propagate the isolates, a single pustule (uredinium) was used to inoculate the susceptible wheat cultivar Boolani in greenhouse..."
Given my many years of experience with P. triticina, it is difficult to imagine that isolates from barley, rye and oats will multiply on wheat.In this regard, I cannot give a positive assessment of this work.
Author Response
See the attached response letter

Reviewer 4 Report
This is an interesting manuscript about the genetic variability of local populations in Iran from 2 2010 to 2017 of Puccinia triticina, the causal agent of leaf rust, also known as brown rust, as a major disease of wheat (Triticum aestivum L and T. durum Desf.) globally using SSRs and ISSRs as molecular markers.
The present work was organized logically, and the results obtained were reliable and persuasive. The results are well presented, their interpretation is relevant, and the methods are highly detailed. I would therefore recommend accepting this manuscript as a limitedly focused paper.
Author Response
The Authors tahnk the reviewer for the positive revision of the manuscript.
Kind regards
S. Olga Cacciola
Round 2
Reviewer 2 Report
I'd like to accept this MS in its current shape. Although some analysis methods I hold conserve comments, this is not my study.
Author Response
Dear Reviewer,
on the behalf of the authors, I thank for accepting the manuscript which has been revised according the suggestions received.
Kind regards
Santa Olga Cacciola
